# Dataset Pruning: Reducing Training Data by Examining Generalization Influence

**Shuo Yang**[1]  **Zeke Xie**[2]  **Hanyu Peng**[2]  **Min Xu**[1]  **Mingming Sun**[2]  **Ping Li**[2]

[1]School of Electrical and Data Engineering, University of Technology Sydney
[2]Cognitive Computing Lab, Baidu Research

## Abstract

The great success of deep learning heavily relies on increasingly larger training data, which comes at a price of huge computational and infrastructural costs. This poses crucial questions that, do all training data contribute to model's performance? How much does each individual training sample or a sub-training-set affect the model's generalization, and how to construct the *smallest* subset from the entire training data as a proxy training set without significantly sacrificing the model's performance? To answer these, we propose *dataset pruning*, an optimization-based sample selection method that can (1) examine the influence of removing a particular set of training samples on model's generalization ability with theoretical guarantee, and (2) construct the *smallest subset* of training data that yields strictly constrained generalization gap. The empirically observed generalization gap of dataset pruning is substantially consistent with our theoretical expectations. Furthermore, the proposed method prunes 40% training examples on the CIFAR-10 dataset, halves the convergence time with only 1.3% test accuracy decrease, which is superior to previous score-based sample selection methods.

## 1 Introduction

The great advances in deep learning over the past decades have been powered by ever-bigger models crunching ever-bigger amounts of data. However, this success comes at a price of huge computational and infrastructural costs for network training, network inference, hyper-parameter tuning, and model architecture search. While lots of research efforts seek to reduce the network inference cost by pruning redundant parameters Blalock et al. (2020); Liu et al. (2018); Molchanov et al. (2019), scant attention has been paid to the *data redundant* problem, which is crucial for network training and parameter tuning efficiency.

Investigating the data redundant problem not only helps to improve the training efficiency, but also helps us understand the representation ability of small data and how many training samples is required and sufficient for a learning system. Previous literatures on subset selection try to sort and select a fraction of training data according to a *scalar score* computed based on some criterions, such as the distance to class center Welling (2009); Rebuffi et al. (2017); Castro et al. (2018); Belouadah & Popescu (2020), the distance to other selected examples Wolf (2011); Sener & Savarese (2018b), the forgetting score Toneva et al. (2019a), and the gradient norm Paul et al. (2021). However, these methods are (a) *heuristic* and lack of theoretically guaranteed generalization for the selected data, also (b) discard the *joint effect*, *i.e.,* the norm of *averaged* gradient vector of two *high-gradient-norm* samples could be zero if the direction of these two samples' gradient is opposite.

To go beyond these limitations, a natural ask is, how much does *a combination of particular training examples* contribute to the model's generalization ability? However, simply evaluating the test performance drop caused by removing each possible subset is not acceptable. Because it requires to re-train the model for $2^n$ times given a dataset with size $n$. Therefore, the key challenges of dataset pruning are: (1) *how to efficiently estimate the generalization influence of all possible subsets without iteratively re-training the model*, and (2) *how to identify the smallest subset of the original training data with strictly constrained generalization gap*.

In this work, we present an optimization-based dataset pruning method that can identify the *largest* redundant subset from the entire training dataset with (a) theoretically guaranteed generalization gap

and (b) consideration of the *joint influence* of all collected data. Specifically, we define the *parameter influence* of a training example as the model's parameter change caused by omitting the example from training. The parameter influence can be linearly approximated without re-training the model by Influence Function Koh & Liang (2017a). Then, we formulate the subset selection as a *constrained discrete optimization* problem with an objective of maximizing the number of collected samples, and a constraint of penalizing the network parameter change caused by removing the collected subset within a given threshold $\epsilon$. We then conduct extensive theoretical and empirical analysis to show that the generalization gap of the proposed dataset pruning can be upper bounded by the pre-defined threshold $\epsilon$. Superior to previous score-based sample selection methods, our proposed method prunes 40% training examples in CIFAR-10 Krizhevsky (2009), halves the convergence time, and achieves only 1.3% test accuracy drop. Before delving into details, we summarize our contributions as below:

- This paper proposes a *dataset pruning* problem, which extends the sample selection problem from *cardinality* constraint to *generalization* constraint. Specifically, previous sample selection methods find a data subset with *fixed size budget* and try to improve their performance, while dataset pruning tries to identify the *smallest subset* that satisfies the expected generalization ability.

- This paper proposes to leverage the influence function to approximate the network parameter change caused by omitting each individual training example. Then an optimization-based method is proposed to identify the largest subset that satisfies the expected parameter change.

- We prove that the generalization gap caused by removing the identified subset can be up-bounded by the network parameter change that was strictly constrained during the dataset pruning procedure. The observed empirical result is substantially consistent with our theoretical expectation.

- The experimental results on dataset pruning and neural architecture search (NAS) demonstrate that the proposed dataset pruning method is extremely effective on improving the network training and architecture search efficiency.

The rest of this paper is organized as follows. In Section. 2, we briefly review existing sample selection and dataset condensation research, where we will discuss the major differences between our proposed dataset pruning and previous methods. In Section. 3, we present the formal definition of the dataset pruning problem. In Section. 4 and Section. 5, our optimization-based dataset pruning method and its generalization bound are introduced. In Section. 6, we conduct extensive experiments to verify the validity of the theoretical result and the effectiveness of dataset pruning. Finally, in Section. 7, we conclude our paper.

## 2 RELATED WORKS

Dataset pruning is orthogonal to few-shot learning Finn et al. (2017); Vinyals et al. (2016); Snell et al. (2017); Chen et al. (2019); Zhang et al. (2020); Yang et al. (2021a;b). Few-shot learning aims at improving the performance given limited training data, while dataset pruning aims at reducing the training data without hurting the performance much.

Dataset pruning is closely related to the data selection methods, which try to identify the most representative training samples. Classical data selection methods Agarwal et al. (2004); Har-Peled & Mazumdar (2004); Feldman et al. (2013) focus on clustering problems. Recently, more and more data selection methods have been proposed in continual learning Rebuffi et al. (2017); Toneva et al. (2019b); Castro et al. (2018); Aljundi et al. (2019) and active learning Sener & Savarese (2018a) to identify which example needs to be stored or labeled. The data selection methods typically rely on a pre-defined criterion to compute a *scalar score* for each training example, *e.g.* the compactness Rebuffi et al. (2017); Castro et al. (2018), diversity Sener & Savarese (2018b); Aljundi et al. (2019), forgetfulness Toneva et al. (2019b), or the gradient norm Paul et al. (2021), then rank and select the training data according to the computed score. However, these methods are heuristic and lack of generalization guarantee, they also discard the influence interaction between the collected samples. Our proposed dataset pruning method overcomes these shortcomings.

Another line of works on reducing the training data is dataset distillation Wang et al. (2018); Such et al. (2020); Sucholutsky & Schonlau (2019); Bohdal et al. (2020); Nguyen et al. (2021a;b); Cazenavette et al. (2022) or dataset condensation Zhao et al. (2021); Zhao & Bilen (2021a;b); Wang et al. (2022);

Jin et al. (2021; 2022). This series of works focus on *synthesizing* a small but informative dataset as an alternative to the original large dataset. For example, Dataset Distillation Wang et al. (2018) try to learn the synthetic dataset by directly minimizing the classification loss on the real training data of the neural networks trained on the synthetic data. Cazenavette et al. (2022) proposed to learn the synthetic images by matching the training trajectories. Zhao et al. (2021); Zhao & Bilen (2021a;b) proposed to learn the synthetic images by matching the gradients and features. However, due to the computational power limitation, these methods usually only synthesize an extremely small number of examples (*e.g.* 50 images per class) and the performance is far from satisfactory. Therefore, the performances of dataset distillation and dataset pruning are not directly comparable.

Our method is inspired by Influence Function Koh & Liang (2017a) in statistical machine learning. Removing a training example from the training dataset and not damage the generalization indicates that the example has a small influence on the expected test loss. Earlier works focused on studying the influence of removing training points from linear models, and later works extended this to more general models Hampel et al. (2011); Cook (1986); Thomas & Cook (1990); Chatterjee & Hadi (1986); Wei et al. (1998). Liu et al. (2014) used influence functions to study model robustness and to fo fast cross-validation in kernel methods. Kabra et al. (2015) defined a different notion of influence that is specialized to finite hypothesis classes. Koh & Liang (2017a) studied the influence of weighting an example on model's parameters. Borsos et al. (2020) proposes to construct a coreset by learning a per-sample weight score, the proposed selection rule eventually converge to the formulation of influence function Koh & Liang (2017b). But different with our work, Borsos et al. (2020) did not directly leverage the influence function to select examples and cannot guarantee the generalization of the selected examples.

## 3 PROBLEM DEFINITION

Given a large-scale dataset $\mathcal{D} = \{z_1, \ldots, z_n\}$ containing $n$ training points where $z_i = (x_i, y_i) \in \mathcal{X} \times \mathcal{Y}$, $\mathcal{X}$ is the input space and $\mathcal{Y}$ is the label space. The goal of dataset pruning is to identify a set of redundant training samples from $\mathcal{D}$ as many as possible and remove them to reduce the training cost. The identified redundant subset, $\hat{\mathcal{D}} = \{\hat{z}_1, \ldots, \hat{z}_m\}$ and $\hat{\mathcal{D}} \subset \mathcal{D}$, should have a minimal impact on the learned model, *i.e.* the test performances of the models learned on the training sets before and after pruning should be very close, as described below:

$$\mathbb{E}_{z \sim P(\mathcal{D})} L(z, \hat{\theta}) \simeq \mathbb{E}_{z \sim P(\mathcal{D})} L(z, \hat{\theta}_{-\hat{\mathcal{D}}}) \tag{1}$$

where $P(\mathcal{D})$ is the data distribution, $L$ is the loss function, and $\hat{\theta}$ and $\hat{\theta}_{-\hat{\mathcal{D}}}$ are the empirical risk minimizers on the training set $\mathcal{D}$ before and after pruning $\hat{\mathcal{D}}$, respectively, *i.e.*, $\hat{\theta} = \arg\min_{\theta \in \Theta} \frac{1}{n} \sum_{z_i \in \mathcal{D}} L(z_i, \theta)$ and $\hat{\theta}_{-\hat{\mathcal{D}}} = \arg\min_{\theta \in \Theta} \frac{1}{n-m} \sum_{z_i \in \mathcal{D} \setminus \hat{\mathcal{D}}} L(z_i, \theta)$. Considering the neural network is a locally smooth function Rifai et al. (2012); Goodfellow et al. (2014); Zhao et al. (2021), similar weights ($\theta \approx \hat{\theta}$) imply similar mappings in a local neighborhood and thus generalization performance. Therefore, we can achieve Eq. 1 by obtaining a $\hat{\theta}_{-\hat{\mathcal{D}}}$ that is very close to $\hat{\theta}$ (the distance between $\hat{\theta}_{-\hat{\mathcal{D}}}$ and $\hat{\theta}$ is smaller than a given very small value $\epsilon$). To this end, we first define the dataset pruning problem on the perspective of model parameter change, we will later provide the theoretical evidence that the generalization gap in Eq. 1 can be upper bounded by the parameter change in Section. 5.

**Definition 1** ($\epsilon$-redundant subset.). *Given a dataset $\mathcal{D} = \{z_1, \ldots, z_n\}$ containing $n$ training points where $z_i = (x_i, y_i) \in \mathcal{X} \times \mathcal{Y}$, considering $\hat{\mathcal{D}}$ is a subset of $\mathcal{D}$ where $\hat{\mathcal{D}} = \{\hat{z}_1, \ldots, \hat{z}_m\}$, $\hat{\mathcal{D}} \subset \mathcal{D}$. We say $\hat{\mathcal{D}}$ is an $\epsilon$-redundant subset of $\mathcal{D}$ if $\left\| \hat{\theta}_{-\hat{\mathcal{D}}} - \hat{\theta} \right\|_2 \leq \epsilon$, where $\hat{\theta}_{-\hat{\mathcal{D}}} = \arg\min_{\theta \in \Theta} \frac{1}{n-m} \sum_{z_i \in \mathcal{D} \setminus \hat{\mathcal{D}}} L(z_i, \theta)$ and $\hat{\theta} = \arg\min_{\theta \in \Theta} \frac{1}{n} \sum_{z_i \in \mathcal{D}} L(z_i, \theta)$, then write $\hat{\mathcal{D}}$ as $\hat{\mathcal{D}}_\epsilon$.*

**Dataset Pruning:** Given a dataset $\mathcal{D}$, dataset pruning aims at finding its *largest* $\epsilon$-redundant subset $\hat{\mathcal{D}}_\epsilon^{max}$, *i.e.*, $\forall \hat{\mathcal{D}}_\epsilon \subset \mathcal{D}, |\hat{\mathcal{D}}_\epsilon^{max}| \geq |\hat{\mathcal{D}}_\epsilon|$, so that the pruned dataset can be constructed as $\tilde{\mathcal{D}} = \mathcal{D} \setminus \hat{\mathcal{D}}_\epsilon^{max}$.

## 4 METHOD

To achieve the goal of dataset pruning, we need to evaluate the model's parameter change $\left\| \hat{\theta}_{-\hat{\mathcal{D}}} - \hat{\theta} \right\|_2$ caused by removing each possible subset of $\mathcal{D}$. However, it is impossible to re-train the model for $2^n$ times to obtain $\hat{\mathcal{D}}_\epsilon^{max}$ for a given dataset $\mathcal{D}$ with size $n$. In this section, we propose to efficiently approximate the $\hat{\mathcal{D}}_\epsilon^{max}$ without the need to re-train the model.

### 4.1 PARAMETER INFLUENCE ESTIMATION

We start from studying the model parameter change of removing each single training sample $z$ from the training set $\mathcal{D}$. The change can be formally written as $\hat{\theta}_{-z} - \hat{\theta}$, where $\hat{\theta}_{-z} = \arg\min_{\theta \in \Theta} \frac{1}{n-1} \sum_{z_i \in \mathcal{D}, z_i \neq z} L(z_i, \theta)$. Estimating the parameter change for each training example by re-training the model for $n$ times is also unacceptable time-consuming, because $n$ is usually on the order of tens or even hundreds of thousands.

Alternatively, the researches of Influence Function Cook (1977); Cook & Weisberg (1980); Cook (1986); Cook & Weisberg (1982); Koh & Liang (2017a) provide us an accurate and fast estimation of the parameter change caused by weighting an example $z$ for training. Considering a training example $z$ was weighted by a small $\delta$ during training, the empirical risk minimizer can be written as $\hat{\theta}_{\delta,z} = \arg\min_{\theta \in \Theta} \frac{1}{n} \sum_{z_i \in \mathcal{D}} L(z_i, \theta) + \delta L(z, \theta)$. Assigning $-\frac{1}{n}$ to $\delta$ is equivalent to removing the training example $z$. Then, the influence of weighting $z$ on the parameters is given by

$$\mathcal{I}_{\text{param}}(z) = \left. \frac{\mathrm{d}\hat{\theta}_{\delta,z}}{\mathrm{d}\delta} \right|_{\delta=0} = -H_{\hat{\theta}}^{-1} \nabla_\theta L(z, \hat{\theta}) \tag{2}$$

where $H_{\hat{\theta}} = \frac{1}{n} \sum_{z_i \in \mathcal{D}} \nabla_\theta^2 L(z_i, \hat{\theta})$ is the Hessian and positive definite by assumption, $\mathcal{I}_{\text{param}}(z) \in \mathbb{R}^N$, $N$ is the number of network parameters. The proof of Eq.equation 2 can be found in Koh & Liang (2017a). Then, we can linearly approximate the parameter change due to removing $z$ without retraining the model by computing $\hat{\theta}_{-z} - \hat{\theta} \approx -\frac{1}{n} \mathcal{I}_{\text{param}}(z) = \frac{1}{n} H_{\hat{\theta}}^{-1} \nabla_\theta L(z, \hat{\theta})$. Similarly, we can approximate the parameter change caused by removing a subset $\hat{\mathcal{D}}$ by accumulating the parameter change of removing each example, $\hat{\theta}_{-\hat{\mathcal{D}}} - \hat{\theta} \approx \sum_{z_i \in \hat{\mathcal{D}}} -\frac{1}{n} \mathcal{I}_{\text{param}}(z_i) = \sum_{z_i \in \hat{\mathcal{D}}} \frac{1}{n} H_{\hat{\theta}}^{-1} \nabla_\theta L(z_i, \hat{\theta})$. The accumulated individual influence can accurately reflect the influence of removing a group of data, as proved in Koh et al. (2019).

### 4.2 DATASET PRUNING AS DISCRETE OPTIMIZATION

Combining definition 1 and the parameter influence function (Eq. equation 2), it is easy to derive that if the parameter influence of a subset $\hat{\mathcal{D}}$ satisfies $\left\| \sum_{z_i \in \hat{\mathcal{D}}} -\frac{1}{n} \mathcal{I}_{\text{param}}(z_i) \right\|_2 \leq \epsilon$, then it is an $\epsilon$-redundant subset of $\mathcal{D}$. Denote $\mathbb{S} = \{-\frac{1}{n}\mathcal{I}_{\text{param}}(z_1), \cdots, -\frac{1}{n}\mathcal{I}_{\text{param}}(z_n)\}$, to find the largest $\epsilon$-redundant subset $\hat{\mathcal{D}}_\epsilon^{max}$ that satisfies the conditions of (1) $\left\| \sum_{z_i \in \hat{\mathcal{D}}_\epsilon^{max}} -\frac{1}{n}\mathcal{I}(z_i) \right\|_2 \leq \epsilon$ and (2) $\forall \hat{\mathcal{D}}_\epsilon \subset \mathcal{D}, |\hat{\mathcal{D}}_\epsilon^{max}| \geq |\hat{\mathcal{D}}_\epsilon|$ simultaneously, we formulate the generalization-guaranteed dataset pruning as a *discrete optimization* problem as below (a):

**(a) generalization-guaranteed pruning:**

$$\begin{aligned} \underset{W}{\text{maximize}} \quad & \sum_{i=1}^n W_i \\ \text{subject to} \quad & \left\| W^T \mathbb{S} \right\|_2 \leq \epsilon \\ & W \in \{0,1\}^n \end{aligned} \tag{3}$$

**(b) cardinality-guaranteed pruning:**

$$\begin{aligned} \underset{W}{\text{minimize}} \quad & \left\| W^T \mathbb{S} \right\|_2 \\ \text{subject to} \quad & \sum_{i=1}^n W_i = m \\ & W \in \{0,1\}^n \end{aligned} \tag{4}$$

where $W$ is a discrete variable that is needed to be optimized. For each dimension $i$, $W_i = 1$ indicates the training sample $z_i$ is selected into $\hat{\mathcal{D}}_\epsilon^{max}$, while $W_i = 0$ indicates the training sample $z_i$ is not pruned. After solving $W$ in Eq .3, the largest $\epsilon$-redundant subset can be constructed as $\hat{\mathcal{D}}_\epsilon^{max} = \{z_i | \forall z_i \in \mathcal{D}, W_i = 1\}$. For some scenarios when we need to specify the removed subset

---

**Algorithm 1** Generalization guaranteed dataset pruning.

---

**Require:** Dataset $\mathcal{D} = \{z_1, \ldots, z_n\}$
**Require:** Random initialized network $\theta$;
**Require:** Expected generalization drop $\epsilon$;
1: $\hat{\theta} = \arg\min_{\theta \in \Theta} \frac{1}{n} \sum_{z_i \in \mathcal{D}} L(z_i, \theta)$;                    //compute ERM on $\mathcal{D}$
2: Initialize $\mathbb{S} = \phi$;
3: **for** $i = 1, 2, \ldots, n$ **do**
4:     $\mathbb{S}_i = -\frac{1}{n}\mathcal{I}_{\text{param}}(z_i) = \frac{1}{n}H_{\hat{\theta}}^{-1}\nabla_\theta L(z_i, \hat{\theta})$;    //store the parameter influence of each example
5: **end for**
6: Initialize $W \in \{0, 1\}^n$;
7: Solve the following problem to get $W$:

$$\begin{aligned} \underset{W}{\text{maximize}} \quad & \sum_{i=1}^{n} W_i && \text{//maximize the subset size} \\ \text{subject to} \quad & \left\| W^T \mathbb{S} \right\|_2 \leq \epsilon && \text{//guarantee the generalization drop} \\ & W \in \{0,1\}^n \end{aligned}$$

8: Construct the largest $\epsilon$-redundant subset $\hat{\mathcal{D}}_\epsilon^{max} = \{z_i | \forall z_i \in \mathcal{D}, W_i = 1\}$;
9: **return** Pruned dataset: $\tilde{\mathcal{D}} = \mathcal{D} \setminus \hat{\mathcal{D}}_\epsilon^{max}$;

---

size $m$ and want to minimize their influence on parameter change, we provide cardinality-guaranteed dataset pruning in Eq. 4.

## 5 GENERALIZATION ANALYSIS

In this section, we theoretically formulate the generalization guarantee of the minimizer given by the pruned dataset. Our theoretical analysis suggests that the proposed dataset pruning method have a relatively tight upper bound on the expected test loss, given a small enough $\epsilon$.

For simplicity, we first assume that we prune only one training sample $z$ (namely, $\delta$ is one-dimensional). Following the classical results, we may write the influence function of the test loss as

$$\mathcal{I}_{\text{loss}}(z_{\text{test}}, z) = \left. \frac{\mathrm{d}L\left(z_{\text{test}}, \hat{\theta}_{z,\delta}\right)}{\mathrm{d}\delta} \right|_{\delta=0} = \nabla_\theta L(z_{\text{test}}, \hat{\theta})^\top \left. \frac{\mathrm{d}\hat{\theta}_{\delta,z}}{\mathrm{d}\delta} \right|_{\delta=0} = \nabla_\theta L(z_{\text{test}}, \hat{\theta})^\top \mathcal{I}_{\text{param}}(z) \tag{5}$$

which indicates the first-order derivative of the test loss $L\left(z_{\text{test}}, \hat{\theta}_{z,\delta}\right)$ with respect to $\delta$.

We may easily generalize the theoretical analysis to the case that prunes $m$ training samples, if we let $\delta$ be a $m$-dimensional vector $(-\frac{1}{n}, \ldots, -\frac{1}{n})$ and $\hat{z} \in \hat{\mathcal{D}}$. Then we may write the multi-dimensional form of the influence function of the test loss as

$$\mathcal{I}_{\text{loss}}(z_{\text{test}}, \hat{\mathcal{D}}) = \nabla_\theta L(z_{\text{test}}, \hat{\theta})^\top \left. \frac{\mathrm{d}\hat{\theta}_{\delta,\hat{z}}}{\mathrm{d}\delta} \right|_{\delta=(0,\ldots,0)} = (\mathcal{I}_{\text{loss}}(z_{\text{test}}, \hat{z}_1), \ldots, \mathcal{I}_{\text{loss}}(z_{\text{test}}, \hat{z}_m)), \tag{6}$$

which is an $N \times m$ matrix and $N$ indicates the number of parameters.

We define the expected test loss over the data distribution $P(\mathcal{D})$ as $\mathcal{L}(\theta) = \mathbb{E}_{z_{\text{test}} \sim P(\mathcal{D})} L(z_{\text{test}}, \theta)$ and define the generalization gap due to dataset pruning as $|\mathcal{L}(\hat{\theta}_{-\hat{\mathcal{D}}}) - \mathcal{L}(\hat{\theta})|$. By using the influence function of the test loss Singh et al. (2021), we obtain Theorem 1 which formulates the upper bound of the generalization gap.

**Theorem 1** (Generalization Gap of Dataset Pruning). *Suppose that the original dataset is $\mathcal{D}$ and the pruned dataset is $\hat{\mathcal{D}} = \{\hat{z}_i\}_{i=1}^m$. If $\left\| \sum_{\hat{z}_i \in \hat{\mathcal{D}}} \mathcal{I}_{\text{param}}(\hat{z}_i) \right\|_2 \leq \epsilon$, we have the upper bound of the*

*generalization gap as*

$$\sup |\mathcal{L}(\hat{\theta}_{-\hat{\mathcal{D}}}) - \mathcal{L}(\hat{\theta})| = \mathcal{O}(\frac{\epsilon}{n} + \frac{m}{n^2}). \tag{7}$$

*Proof.* We express the test loss at $\hat{\theta}_{-\hat{\mathcal{D}}}$ using the first-order Taylor approximation as

$$L\left(z_{\text{test}}, f(\hat{\theta}_{-\hat{\mathcal{D}}})\right) = L\left(z_{\text{test}}, f(\hat{\theta})\right) + \mathcal{I}_{\text{loss}}(z_{\text{test}}, \hat{\mathcal{D}})\delta + \mathcal{O}(\|\delta\|^2), \tag{8}$$

where the last term $\mathcal{O}(\|\delta\|^2)$ is usually ignorable in practice, because $\|\delta\|^2 = \frac{m}{n^2}$ is very small for popular benchmark datasets. The same approximation is also used in related papers which used influence function for generalization analysis Singh et al. (2021). According to Equation equation 8, we have

$$\mathcal{L}(\hat{\theta}_{-\hat{\mathcal{D}}}) - \mathcal{L}(\hat{\theta}) \approx \mathbb{E}_{z_{\text{test}} \sim P(\mathcal{D})}[\mathcal{I}_{\text{loss}}(z_{\text{test}}, \hat{\mathcal{D}})]\delta \tag{9}$$

$$= \sum_{\hat{z}_i \in \hat{\mathcal{D}}} -\frac{1}{n}\mathbb{E}_{z_{\text{test}} \sim P(\mathcal{D})}[\mathcal{I}_{\text{loss}}(z_{\text{test}}, \hat{z}_i)] \tag{10}$$

$$= -\frac{1}{n}\mathbb{E}_{z_{\text{test}} \sim P(\mathcal{D})}[-\nabla_\theta L(z_{\text{test}}, \hat{\theta})^\top] \left[\sum_{\hat{z}_i \in \hat{\mathcal{D}}} H_{\hat{\theta}}^{-1} \nabla_\theta L(\hat{z}_i, \hat{\theta})\right] \tag{11}$$

$$\leq \frac{1}{n}\|\nabla_\theta \mathcal{L}(\hat{\theta})\|_2 \left\|\sum_{\hat{z}_i \in \hat{\mathcal{D}}} \mathcal{I}_{\text{param}}(\hat{z}_i)\right\|_2 \tag{12}$$

$$\leq \frac{\epsilon}{n}\|\nabla_\theta \mathcal{L}(\hat{\theta})\|_2, \tag{13}$$

where the first inequality is the Cauchy–Schwarz inequality, the second inequality is based on the algorithmic guarantee that $\left\|\sum_{\hat{z}_i \in \hat{\mathcal{D}}} \mathcal{I}_{\text{param}}(\hat{z}_i)\right\|_2 \leq \epsilon$ in Eq. equation 3, and $\epsilon$ is a hyperparameter. Given the second-order Taylor term, finally, we obtain the upper bound of the expected loss as

$$\sup |\mathcal{L}(\hat{\theta}_{-\hat{\mathcal{D}}}) - \mathcal{L}(\hat{\theta})| = \mathcal{O}(\frac{\epsilon}{n} + \frac{m}{n^2}). \tag{14}$$

The proof is complete. □

Theorem 1 demonstrates that, if we hope to effectively decrease the upper bound of the generalization gap due to dataset pruning, we should focus on constraining or even directly minimizing $\epsilon$. The basic idea of the proposed optimization-based dataset pruning method exactly aims at penalizing $\epsilon$ via discrete optimization, shown in Eq. 3 and Eq. 4.

Moreover, our empirical results in the following section successfully verify the estimated generalization gap. The estimated generalization gap of the proposed optimization-based dataset pruning approximately has the order of magnitude as $\mathcal{O}(10^{-3})$ on CIFAR-10 and CIFAR-100, which is significantly smaller the estimated generalization gap of random dataset pruning by more than one order of magnitude.

## 6 EXPERIMENT

In the following paragraph, we conduct experiments to verify the validity of the theoretical results and the effectiveness of the proposed dataset pruning method. In Section. 6.1, we introduce all experimental details. In Section. 6.2, we empirically verify the validity of the Theorem. 1. In Section. 6.3 and Section. 6.4, we compare our method with several baseline methods on dataset pruning performance and cross-architecture generalization. Finally, in Section. 6.5, we show the proposed dataset pruning method is extremely effective on improving the training efficiency.

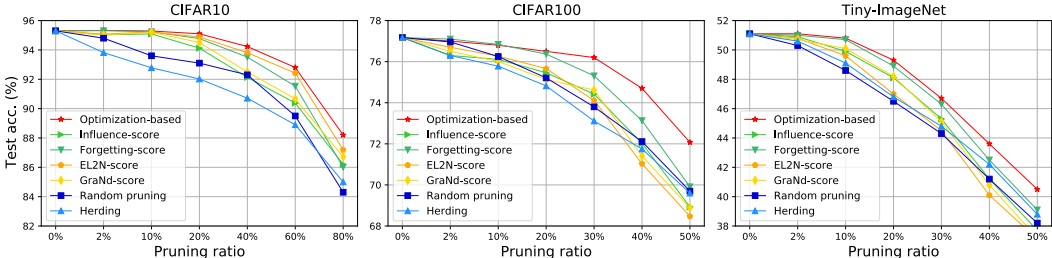

Figure 1: We compare our proposed optimization-based dataset pruning method with several sample-selection baselines. Our optimization-based pruning method considers the 'group effect' of pruned examples and exhibits superior performance, especially when the pruning ratio is high.

## 6.1 EXPERIMENT SETUP AND IMPLEMENTATION DETAILS

We evaluate dataset pruning methods on CIFAR10, CIFAR100 Krizhevsky (2009), and TinyImageNet Le & Yang (2015) datasets. CIFAR10 and CIFAR100 contains 50,000 training examples from 10 and 100 categories, respectively. TinyImageNet is a larger dataset contains 100,000 training examples from 200 categories. We verify the cross-architecture generalization of the pruned dataset on three popular deep neural networks with different complexity, *e.g.,* SqueezeNet Hu et al. (2018) (1.25M parameters), ResNet18 He et al. (2016) (11.69M parameters), and ResNet50 He et al. (2016) (25.56M parameters). All hyper-parameters and experimental settings of training before and after dataset pruning were controlled to be the same. Specifically, in all experiments, we train the model for 200 epochs with a batch size of 128, a learning rate of 0.01 with cosine annealing learning rate decay strategy, SGD optimizer with the momentum of 0.9 and weight decay of 5e-4, data augmentation of random crop and random horizontal flip. In Eq. 2, we calculate the Hessian matrix inverse by using seconder-order optimization trick Agarwal et al. (2016) which can significantly boost the Hessian matrix inverse estimation time. To further improve the estimation efficiency, we only calculate parameter influence in Eq. 2 for the last linear layer. In Eq. 3 and Eq. 4, we solve the discrete optimization problem using simulated annealing Van Laarhoven & Aarts (1987), which is a popular heuristic algorithm for solving complex discrete optimization problem in a given time budget.

## 6.2 THEORETICAL ANALYSIS VERIFICATION

Our proposed optimization-based dataset pruning method tries to collect the smallest subset by constraining the parameter influence $\epsilon$. In Theorem. 1, we demonstrate that the generalization gap of dataset pruning can be upper-bounded by $\mathcal{O}(\frac{\epsilon}{n} + \frac{m}{n^2})$. The term of $\frac{m}{n^2}$ can be simply ignored since it usually has a much smaller magnitude than $\frac{\epsilon}{n}$. To verify the validity of the generalization guarantee of dataset pruning, we compare the empirically observed test loss gap before and after dataset pruning $\mathcal{L}(\hat{\theta}_{-\hat{\mathcal{D}}}) - \mathcal{L}(\hat{\theta})$ and our theoretical expectation $\frac{\epsilon}{n}$ in Fig. 2. It can be clearly observed that the actual generalization gap is highly consistent with our theoretical prediction. We can also observe that there is a strong correlation between the pruned dataset generalization and $\epsilon$, therefore Eq. 3 effectively guarantees the generalization of dataset pruning by constraining $\epsilon$. Compared with random pruning, our proposed optimization-based dataset pruning exhibits much smaller $\epsilon$ and better generalization.

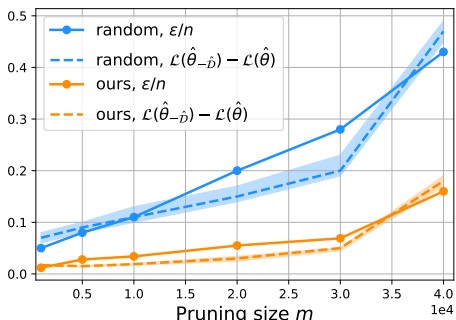

Figure 2: The comparison of empirically observed generalization gap and our theoretical expectation in Theorem. 1. We ignore the term of $m/n^2$ since it has much smaller magnitude with $\epsilon/n$.

## 6.3 DATASET PRUNING

In the previous section, we motivated the optimization-based dataset pruning method by constraining or directly minimizing the network parameter influence of a selected data subset. The theoretical

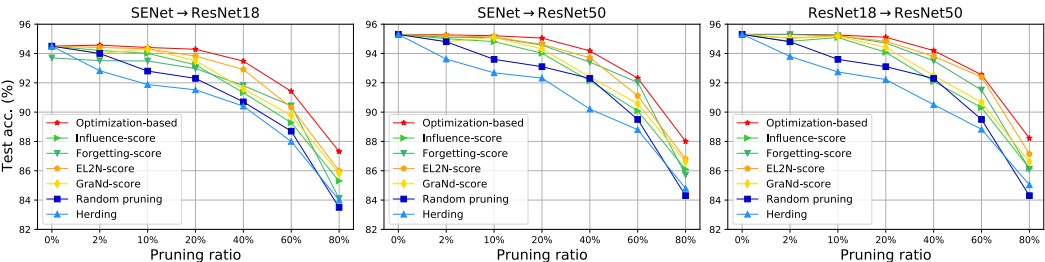

Figure 3: To evaluate the unseen-architecture generalization of the pruned dataset, we prune the CIFAR10 dataset using a relatively small network and then train a larger network on the pruned dataset. We consider three networks from two families with different parameter complexity, SENet (1.25M parameters), ResNet18 (11.69M parameters), and ResNet50 (25.56M parameters). The results indicate that the dataset pruned by small networks can generalize well to large networks.

result and the empirical evidence show that constraining parameter influence can effectively bound the generalization gap of the pruned dataset. In this section, we evaluate the proposed optimization-based dataset pruning method empirically. We show that the test accuracy of a network trained on the pruned dataset is comparable to the test accuracy of the network trained on the whole dataset, and is competitive with other baseline methods.

We prune CIFAR10, CIFAR100 Krizhevsky (2009) and TinyImageNet Le & Yang (2015) using a random initialized ResNet50 network He et al. (2016). We compare our proposed method with the following baselines, (a) Random pruning, which selects an expected number of training examples randomly. (b) Herding Welling (2009), which selects an expected number of training examples that are closest to the cluster center of each class. (c) Forgetting Toneva et al. (2019b), which selects training examples that are easy to be forgotten. (d) GraNd Paul et al. (2021), which selects training examples with larger loss gradient norms. (e) EL2N Paul et al. (2021), which selects training examples with larger norms of the error vector that is the predicted class probabilities minus one-hot label encoding. (f) Influence-score Koh & Liang (2017a), which selects training examples with high norms of the influence function in Eq. 2 without considering the 'group effect'. To make our method comparable to those cardinality-based pruning baselines, we pruned datasets using the cardinality-guaranteed dataset pruning as in Eq. 4. After dataset pruning and selecting a training subset, we obtain the test accuracy by retraining a new random initialized ResNet50 network on only the pruned dataset. The retraining settings of all methods are controlled to be the same. The experimental results are shown in Fig. 1. In Fig. 1, our method consistently surpasses all baseline methods. Among all baseline methods, the forgetting and EL2N methods achieve very close performance to ours when the pruning ratio is small, while the performance gap increases along with the increase of the pruning ratio. The influence-score, as a direct baseline of our method, also performs poorly compared to our optimization-based method. This is because all these baseline methods are *score-based*, they prune training examples by removing the *lowest-score* examples without considering the *influence iteractions* between *high-score examples* and *low-score examples*. The influence of a combination of *high-score examples* may be minor, and vice versa. Our method overcomes this issue by considering the influence of a subset rather than each individual example, by a designed optimization process. The consideration of the group effect make our method outperforms baselines methods, especially when the pruning ratio is high.

## 6.4 UNSEEN ARCHITECTURE GENERALIZATION

We conduct experiments to verify the pruned dataset can generalize well to those unknown or larger network architectures that are inaccessible during dataset pruning. To this end, we use a smaller network to prune the dataset and further use the pruned dataset to train larger network architectures. As shown in Fig. 3, we evaluate the CIFAR10 pruned by SENet Hu et al. (2018) on two unknown and larger architectures, *i.e.*, ResNet18 He et al. (2016) and ResNet50 He et al. (2016). SENet contains 1.25 million parameters, ResNet18 contains 11.69 parameters, and ResNet50 contains 25.56 parameters. The experimental results show that the pruned dataset has a good generalization on network architectures that are unknown during dataset pruning. This indicates that the pruned dataset can be used in a wide range of applications regardless of specific network architecture. In addition, the results also indicate that we can prune the dataset using a small network and the pruned dataset

can generalize to larger network architectures well. These two conclusions make dataset pruning a proper technique to reduce the time consumption of neural architecture search (NAS).

## 6.5 DATASET PRUNING IMPROVES THE TRAINING EFFICIENCY.

The pruned dataset can significantly improve the training efficiency while maintaining the performance, as shown in Fig. 4. Therefore, the proposed dataset pruning benefits when one needs to train many trails on the same dataset. One such application is neural network search (NAS) Zoph et al. (2018) which aims at searching a network architecture that can achieve the best performance for a specific dataset. A potential powerful tool to accelerate the NAS is by searching architectures on the smaller pruned dataset, if the pruned dataset has the same ability of identifying the best network to the original dataset.

We construct a 720 ConvNets searching space with different depth, width, pooling, activation and normalization layers. We train all these 720 models on the whole CIFAR10 training set and four smaller proxy datasets that are constructed by *random*, *herding* Welling (2009), *forgetting* Toneva et al. (2019b), and our proposed dataset pruning method. All the four proxy datasets contain only 100 images per class. We train all models with 100 epochs. The proxy dataset contains 1000 images in total, which occupies 2% storage cost than training on the whole dataset.

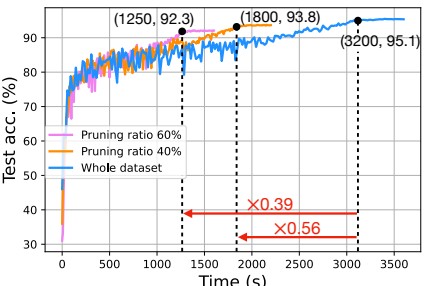

Figure 4: Dataset pruning significantly improves the training efficiency with minor performance scarification. When pruning 40% training examples, the convergence time is nearly halved with only 1.3% test accuracy drop. The pruned dataset can be used to tune hyperparameters and network architectures to reduce the searching time.

Table.1 reports (a) the average test performance of the best selected architectures trained on the whole dataset, (b) the Spearmen's rank correlation coefficient between the validation accuracies obtained by training the selected top 10 models on the proxy dataset and the whole dataset, (c) time of training 720 architectures on a Tesla V100 GPU, and (d) the memory cost. Table. 1 shows that searching 720 architectures on the whole dataset raises huge timing cost. Randomly selecting 2% dataset to perform the architecture search can decrease the searching time from 3029 minutes to 113 minutes, but the searched architecture achieves much lower performance when trained on the whole dataset, making it far from *the best architecture*. Compared with baselines, our proposed dataset pruning method achieves the best performance (the closest one to that of the whole dataset) and significantly reduces the searching time (3% of the whole dataset).

|  | Random | Herding | Forgetting | Ours | Whole Dataset |
|---|---|---|---|---|---|
| Performance (%) | 79.4 | 80.1 | 82.5 | **85.7** | 85.9 |
| Correlation | 0.21 | 0.23 | 0.79 | **0.94** | 1.00 |
| Time cost (min) | **113** | **113** | **113** | **113** | 3029 |
| Storage (imgs) | $10^3$ | $10^3$ | $10^3$ | $10^3$ | $5 \times 10^4$ |

Table 1: Neural architecture search on proxy-sets and whole dataset. The search space is 720 ConvNets. We do experiments on CIFAR10 with 100 images/class proxy dataset selected by random, herding, forgetting, and our proposed optimization-based dataset pruning. The network architecture selected by our pruned dataset achieves very close performance to the upper-bound.

## 7 CONCLUSION

This paper proposes a problem of dataset pruning, which aims at removing redundant training examples with minor impact on model's performance. By theoretically examining the influence of removing a particular subset of training examples on network's parameter, this paper proposes to model the sample selection procedure as a *constrained discrete optimization* problem. During the sample selection, we constrain the network parameter change while maximize the number of collected samples. The collected training examples can then be removed from the training set. The extensive theoretical and empirical studies demonstrate that the proposed optimization-based dataset pruning method is extremely effective on improving the training efficiency while maintaining the model's generalization ability.

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
