# OpenReview forum: "Dataset Pruning: Reducing Training Data by Examining Generalization Influence"
_ICLR.cc/2023/Conference — ICLR 2023 poster_

### Official Review · Reviewer_HgbD · 2022-10-22

**Confidence:** 5
**Correctness:** 4
**Technical Novelty And Significance:** 2
**Empirical Novelty And Significance:** 2
**Recommendation:** 5

**Clarity, Quality, Novelty And Reproducibility:**

The paper is clearly written and the idea is well addressed with empirical support. The novelty is slightly limited since the influence function has been widely considered in dataset selection. The main idea is on formulating the discrete optimization problem based on the influence function, but the computational cost of this optimization could be the main bottleneck in using this method in practice.

**Strength And Weaknesses:**

Strength

- The authors propose a new discrete optimization problem to find a smallest subset of data samples with which a model can be trained with a slight parameter change compared to the model trained with the full dataset. The solution of the proposed optimization can guarantee a small generalization gap.

- The authors provide simulation results demonstrating that the solution of their optimization (subsets of data samples selected for CIFAR-10 and CIFAR-100, resp.) are indeed effective in applications of data pruning and neural network search. Moreover, the theoretical analysis on generalization gap with respect to the pruning size is shown to be quite close to the empirical generalization gap in Figure 2.

Weakness

- Computational complexity of the proposed optimization: Even though the proposed discrete optimization does not require re-training of a model, still the optimization itself may require high computational complexity, increasing exponentially in the number of data samples, since the optimization space for (3) or (4) increases exponentially in $n$, e.g. if m=0.5n in (4) (finding 50\% most informative samples) it require ${n\choose n/2}=\Theta(2^n)$ complexity. Can the authors explain the issue with computational complexity?

- Empirical results: In data pruning experiment shown in Fig.1, the authors did not include the most recent SOTA baselines e.g., the Gradient Normed (GraNd) and the Error L2-Norm (EL2N) scores from Paul et al (2021), where the empirical result shows that for CIFAR-10 dataset 50\% training examples can be removed without affecting the test accuracy. The performance of these baselines might be superior than the authors’ method since the authors claim that their method can prune 40\% of examples with 1.3\% test accuracy drop. Can the authors add this comparison as well?


**Summary Of The Paper:**

This paper considers a problem of selecting a largest redundant subset in training examples, without which the model parameter changes within $\epsilon$ bound compared to the case of training with the full dataset. The applications of such data selection include the data pruning, where the goal is to identify a set of redundant training samples as many as possible and remove them to reduce the training cost, and neural network search, which aims at searching a network architecture that can achieve the best performance for a specific dataset, with a few training samples as possible. The main idea is to use influence function Koh & Liang (2017a) and formulate the discrete optimization problem to find the largest redundant subset of data with the constraint on the change of the model parameters. By empirical results, the authors show that the proposed optimization can find a subset of training samples effective in data pruning and neural network search.


**Summary Of The Review:**

The paper is clearly written and the idea is well addressed with empirical support. The novelty is slightly limited since the influence function has been widely considered in dataset selection. The main idea is on formulating the discrete optimization problem based on the influence function, but the computational cost of this optimization could be the main bottleneck in using this method in practice.

---

> ### Author Response · Authors · 2022-11-13
> **To Reviewer HgbD**
>
> Response to Reviewer HgbD:
>
> Thanks for taking the time to review our paper! We are happy to respond to your comments and questions as below:
>
> ### **Weakness 1: Computational complexity**
>
> Thanks for your great question! The discrete optimization problem in Eq.4 is an NP-hard problem, and it is very time-consuming to get the 'exact solution' (time complexity up to $O(2^n)$). Considering the computational efficiency, we solve the optimization problem in Eq.4 using a simulated annealing algorithm. Simulated annealing is an efficient algorithm that can find an approximated minima in a **given time budget**. From the table, we can see that the performance of simulated annealing is a good trade-off between the optimal performance (exact solution) and the time consumption.
>
> CIFAR10:
> | Pruning ratio |  10|20| 30| 40| 50|
> |  ----  | ----  | ----  | ----  | ----  | ----  |
> | Exact solution (branch and bound) | 95.31(3mins) | 95.08(62mins) |94.36(193mins)|N/A | N/A |
> | Simulated annealing  | 95.30(30mins)| 95.10(30mins)| 94.23(30mins)| 92.81(30mins)| 88.20(30mins)|
> | Simulated annealing  | 95.33(5mins)| 95.03(5mins)| 94.22(5mins)| 92.86(5mins)| 87.50(5mins) |
> | Simulated annealing  |93.66(1mins)| 93.19(1mins)| 92.81(1mins)| 89.50(1mins)| 84.33(1mins) |
>
> TinyImageNet:
> | Pruning ratio |  10|20| 30| 40| 50|
> |  ----  | ----  | ----  | ----  | ----  | ----  |
> | Simulated annealing  | 50.8(60mins)| 49.3(60mins)| 46.7(60mins)| 43.6(60mins)| 40.5(60mins)|
> | Simulated annealing  | 50.7(30mins)| 49.3(30mins)| 46.2(30mins)| 42.5(30mins)| 39.5(30mins)|
> | Simulated annealing  | 49.3(5mins)| 47.6(5mins)| 45.4(5mins)| 43.1(5mins)| 39.8(5mins)|
> | Simulated annealing  | 48.6(1mins)| 46.5(1mins)| 44.3(1mins)| 41.2(1mins)| 38.2(1mins)|
>
> We will include more details, results, and discussions about the discrete optimization problem in the next revision. Our code for solving the optimization problem will also be made public soon.
>
> ### **Weakness 2: More empirical results**
> We have updated the manuscript to include comparison results to more baseline methods (including EL2N and GraNd), please refer to Fig.1 for more details. We use the official code of EL2N and GraNd (https://github.com/mansheej/data_diet) to prune the datasets, and the model training settings are aligned with all other baseline experiments in our paper. Unfortunately, We cannot reproduce the result of EL2N/GraNd that 'prune 50% CIFAR-10 examples without affecting the test accuracy', we also found some concurrent works also failed to produce this result (Fig.2 in [1], Fig.1 in [2]). This might be caused by the difference in training details between EL2N and our paper. We will make our code and pruned datasets public soon.
>
> [1]Data Valuation Without Training of a Model, ICLR 2023 blind submission (https://openreview.net/forum?id=XIzO8zr-WbM)
>
> [2]Data Subset Selection via Machine Teaching, ICLR 2023 blind submission (https://openreview.net/forum?id=tGHi1HFNBx1)
>
> We hope our responses have addressed all of your concerns and questions. If there are any further questions, we are very glad to continue the discussion.

---

> ### Author Response · Authors · 2022-11-22
> **Dear reviewer, are there any further questions here?**
>
> Dear Reviewer HgbD,
>
> Thanks a lot for your efforts in reviewing this paper. We tried our best to address the mentioned concerns. Are there unclear explanations here? We could further clarify them.
>
> Best,
> Authors

---

### Official Review · Reviewer_63py · 2022-10-23

**Confidence:** 4
**Correctness:** 3
**Technical Novelty And Significance:** 3
**Empirical Novelty And Significance:** 3
**Recommendation:** 8

**Clarity, Quality, Novelty And Reproducibility:**

The paper scores high on all four aspects - quality, clarity, novelty and reproducibility. The paper is generally clear, the contributions are novel, the experiments indicate high quality and the algorithm seems simple enough to be reproducible from description.

**Strength And Weaknesses:**

**Strengths**:
- The paper proposes a simple, yet elegant method to perform dataset pruning which is not only a key practical challenge, but can also contribute to a conceptual understanding of the role of particular data points on weights learned by the model. The proposed method seems sound and overcomes limitations of prior methods.

- The broad structure of the paper’s argument is clear and logical. The authors provide an informative presentation of related works and how their work differs from related works, define the problem formulation, present their solution to the problem, and validate their solution empirically, showing that their method outperforms previous methods. The application of this approach to NAS is also very intruiging and presents a strong contribution.

**Weaknesses**: The paper contains typos, awkward sentence structures, and grammatical errors that could benefit from proofreading. Also, the paper does not provide any discussion of the limitations of the method or its assumptions. In addition, there are some questions about the method itself and the interpretation of experimental results (see below). However, these are areas of improvement for the paper, rather than major weaknesses that warrant rejection.

Questions / Suggestions:

- The paper does not provide an understanding of how far off the proposed algorithm is from an oracle algorithm that yields the best data subsets. To this end, it can be helpful to evaluate this approach for small datasets and models, and compare performance against a brute-force oracle which performs an exhaustive search over all data subsets.

- Is there some analysis on the generalization of the proposed approach from a smaller architecture to a larger one in practice? This might make the proposed approach even more practically relevant at least for the case of NAS.

- Section 6.2, Figure 2
    - The experiment is meant to demonstrate that the generalization gap can be bounded by O(e/n + m/(n^2)) (which is approximately O(e/n)). If this is the case, then the dotted lines should be lower than their corresponding solid lines. However, they both seem to match pointwise, despite the rough nature of the upper bound. Is there a re-scaling of the theoretical results in the graph?

    - What dataset and model are used here? Is it CIFAR10 and ResNet50? It could be helpful to test model architectures of varying orders of complexity to see if empirical results consistently match theoretical results across model architectures or if they deviate more from theoretical results as model complexity increases, as presumably the loss surface becomes less non-linear.

- Section 6.5: Data pruning reduces convergence time, but how much time did it take to perform dataset pruning? It seems that to see the true time savings obtained by the method, the comparison should be [dataset pruning time + NAS training times] vs. [NAS on full training set] (rather than [NAS training time on pruned dataset] vs. [NAS training time on full training set]).

- Regarding this statement in the paper "We compare our method with forgetting Toneva et al. (2019b) because it has the strongest empirical performance compared with Borsos et al. (2020); Paul et al. (2021).", it is still relevant to compare against gradient-norm or loss ranking methods, and as such I would strongly encourage the authors to directly compare against such methods instead of relying on such proxy comparison.

**Summary Of The Paper:**

This paper addresses the problem of dataset pruning, which seeks to remove redundant data points from the training set while minimizing model change as much as possible. The method consists of using influence functions to identify the influence of each datapoint, and use a discrete solver to identify which data subset has the least overall influence. The authors derive a rough upper bound for the generalization gap of this method, which are validated experimentally. Empirical results show that the proposed method outperforms competing dataset selection methods such as herding and data forgetting, also show generalization to unseen architectures and improvement in training efficiency.

**Summary Of The Review:**

Accept. I make this recommendation because, overall, the paper addresses a key problem that has both practical and conceptual implications (i.e. how to prune a dataset, which can reduce the computational intensity of model training in practice and help us better understand how models learn from a dataset) in a compelling manner (i.e. proposing a data pruning method that leverages influence functions and discrete optimization).

---

> ### Author Response · Authors · 2022-11-13
> **To Reviewer 63py**
>
> Response to Reviewer 63py:
>
> Thanks for taking the time to review our paper! We feel sorry for the grammatical errors, typos and sentence structures, please allow us some time to go through and proofread the entire paper to make it presented in a higher quality. For other questions, we provide our responses below:
>
> ### **Adding discussion of the limitations**
>
> A major advantage of our proposed optimization-based sample-selection method is that we are the first one that can guarantee the 'group effect' (the joint effect of removing a group of data points). This advantage also brings a limitation to our method, that is, our method takes a relatively long time (to compute the influence function and solve the optimization) and cannot select examples on-the-fly (pruning during training). Our method needs to first go through the entire dataset to compute a vector (estimated parameter change) for each data point, then prune the dataset by a designed discrete optimization process so as to guarantee the 'joint influence' of removing a set of examples.
>
> Considering the group effect of removing the combination of arbitrary examples is rational and effective, but also time-consuming, compared with previous score-based methods. But fortunately, dataset pruning only needs to be performed once, and the pruned dataset can be used for subsequent unlimited rounds of model training, parameter tuning, model architecture search, etc., to reduce computational and storage consumption. More efficient dataset pruning is also a future research direction.
>
> ### **Questions1: Compare with oracle algorithm**
>
> We agree that constructing an 'optimal' subset would be helpful for evaluating the quality of a pruned dataset in this area. But unfortunately, as we discussed in the paper, even for a dataset that contains 20 examples, we need to train a model for $2^{20}$ (1,048,576) times to find the optimal subset. Not to mention that 20 examples are far away from training a reliable model. We will try to figure out a feasible way to construct the oracle algorithm.
>
> ### **Questions2: The generalization from a smaller architecture to a larger one**
>
> Thanks for your great question! We have updated the Fig.3 in the paper to include more analysis of generalization from smaller models to larger models. The experiment settings are, (1) from SENet (1.25 million parameters) to ResNet18 (11.69 million parameters), (2) from SENet (1.25 million parameters) to ResNet50 (25.56 million parameters), and (3) from ResNet18 (11.69 million parameters) to ResNet50 (25.56 million parameters).
>
> ### **Questions3: Regarding the Fig.2**
>
> Yes, we use CIFAR10 and ResNet50 in Fig.2. We did not re-scale the results in the figure, but as we discussed (please refer to Reviewer W84Z-weakness 2), we use a heuristic algorithm (simulated annealing) to get an approximated minima for the optimization process. The simulated annealing algorithm significantly boosts the dataset pruning but it introduces errors to the solution, leading to high loss values in Fig.2. The errors in Fig.2 could also be introduced by the convex assumption of the influence function. We are conducting more experiments to verify the theoretical results on network architectures with different complexity (SENet and ResNet18), the results will be included in the manuscript once finished.
>
> ### **Questions4: The time consumption of dataset pruning**
>
> The time consumption of our dataset pruning method is shown below:
>
> | |CIFAR10 |CIFAR100| TinyImageNet|
> |  ----  | ----  | ----  | ----  |
> | Influence Function (hours)|  3.3 | 3.5 | 6.7 |
> | Optimization (hours)| 0.5 | 1.0 | 1.0|
> |Overall (hours)| 3.8| 4.5| 7.7|
>
> Although dataset pruning, at the current stage, takes a relatively long time (on cifar10, even longer than training a model on full data). But we want to emphasize that (1) Dataset pruning only needs to be performed once, and the pruned dataset can be used for subsequent unlimited rounds of model training, parameter tuning, model architecture search, etc., to reduce computational and storage consumption. (2) Dataset pruning can help the community better understand the generalization ability of a small portion of data. (3) More efficient dataset pruning is also a future research direction.
>
> ### **Questions5: More baseline methods**
> Thanks for your great suggestion! We have updated the manuscript to include results of more baseline methods, please refer to Fig.1 for more details.
>
> We hope our responses have addressed all of your concerns and questions. If there are any further questions, we are very glad to continue the discussion.

---

### Official Review · Reviewer_W84Z · 2022-10-23

**Confidence:** 5
**Correctness:** 4
**Technical Novelty And Significance:** 3
**Empirical Novelty And Significance:** 3
**Recommendation:** 6

**Clarity, Quality, Novelty And Reproducibility:**

- Clarity: The paper is well-written and clear.

- Novelty: The components of the framework is not novel, however the authors combine two components well to make it work for dataset pruning.

- Reproducibility: The results could be reproduced with the information given in the paper.

**Strength And Weaknesses:**

First, I want to highlight that dataset pruning is an important problem and having a robust framework can help in mitigating multiple re-training routines, which can be expensive.  The framework proposed in the paper is new and shows good improvements over other baselines, though the coverage of experiments is slightly limited, which is my main concern.

Strengths:

	- While both influence function and the optimization algorithm is not new, their combination is relatively new.

	- The paper is very well written and is coherent.

	- The method has good improvements over the baselines (forgetting and herding), though would like to see some more baselines such as [1] in the main paper. Moreover, showing the generalizability of the framework to other architectures is critical which the authors have shown positive results on.

Weaknesses:

	- While the paper shows improvements on CIFAR derivatives, it lacks analysis or results on other datasets (e.g., ImageNet derivatives). Verifying the effectiveness of the framework on ImageNet-1k or even ImageNet-100 is important. These results ideally can be presented in the main paper.

	- The authors should add some details on how to solve the optimization in the main paper.  It's an important piece of information currently lacking in the paper.

	- Some baselines such as [1] are not considered and should be added.


I feel that influence function can be replaced by other influence estimation methods such as datamodels[2] or tracin[3]. It will be beneficial to understand if the updated framework results in better pruning than the baselines. I am assuming it would result in better pruning results, however it would be beneficial to understand which influence based methods are particularly suitable for pruning.


[1]. https://arxiv.org/pdf/2107.07075

[2]. https://arxiv.org/abs/2202.00622

[3]. https://arxiv.org/abs/2002.08484


**Summary Of The Paper:**

The paper proposes an optimization-based framework with influence functions for subset selection (or dataset pruning). The authors show in certain cases that the influence based method can be used to prune large datasets with only a small increase in the generalization error. From a practical point of view, this framework can be useful in situations when there is a repeated need for re-training models.

**Summary Of The Review:**

Although the components of the framework is not novel, their combination leads to a generalizable framework (with theoretical guarantees) for dataset pruning.  I will be leaning towards a weak-accept, as I feel that the framework can potentially be used to alleviate the cost of re-training.

---

> ### Author Response · Authors · 2022-11-13
> **To Reviewer W84Z**
>
> Response to Reviewer W84Z:
>
> Thanks for taking the time to review our paper! We are happy to respond to your comments and questions as below:
>
> ### **Weakness 1 & 3**
> Thanks for your great suggestion! We have updated the manuscript to include results on the TinyImageNet (200 classes, 100,000 examples) and comparison to more baseline methods (including [1]), please refer to Fig.1 for more details.
>
> ### **Weakness 2**
> Thanks for your great suggestion! The discrete optimization problem in Eq.4 is an NP-hard problem, and it is very time-consuming to get the 'exact solution'. Considering the computational efficiency, we solve the optimization problem in Eq.4 using a simulated annealing algorithm. Simulated annealing is an efficient algorithm that can find an approximated minima in a given time budget. From the table, we can see that the performance of simulated annealing is a good trade-off between the optimal performance (exact solution) and the time consumption.
>
> CIFAR10:
> | Pruning ratio |  10|20| 30| 40| 50|
> |  ----  | ----  | ----  | ----  | ----  | ----  |
> | Exact solution (branch and bound) | 95.31(3mins) | 95.08(62mins) |94.36(193mins)|N/A | N/A |
> | Simulated annealing  | 95.30(30mins)| 95.10(30mins)| 94.23(30mins)| 92.81(30mins)| 88.20(30mins)|
> | Simulated annealing  | 95.33(5mins)| 95.03(5mins)| 94.22(5mins)| 92.86(5mins)| 87.50(5mins) |
> | Simulated annealing  |93.66(1mins)| 93.19(1mins)| 92.81(1mins)| 89.50(1mins)| 84.33(1mins) |
>
> TinyImageNet:
> | Pruning ratio |  10|20| 30| 40| 50|
> |  ----  | ----  | ----  | ----  | ----  | ----  |
> | Simulated annealing  | 50.8(60mins)| 49.3(60mins)| 46.7(60mins)| 43.6(60mins)| 40.5(60mins)|
> | Simulated annealing  | 50.7(30mins)| 49.3(30mins)| 46.2(30mins)| 42.5(30mins)| 39.5(30mins)|
> | Simulated annealing  | 49.3(5mins)| 47.6(5mins)| 45.4(5mins)| 43.1(5mins)| 39.8(5mins)|
> | Simulated annealing  | 48.6(1mins)| 46.5(1mins)| 44.3(1mins)| 41.2(1mins)| 38.2(1mins)|
>
> We will include more details, results, and discussions about the discrete optimization problem in the next revision. Our code for solving the optimization problem will also be made public soon.
>
>
> We hope our responses have addressed all of your concerns and questions. If there are any further questions, we are very glad to continue the discussion.

---

> > ### Comment · Reviewer_W84Z · 2022-11-15
> > **Reply to Authors**
> >
> > I thank the authors for their response; I maintain my score and inclined towards accepting it.

---

### Official Review · Reviewer_F3Cs · 2022-10-24

**Confidence:** 3
**Correctness:** 3
**Technical Novelty And Significance:** 2
**Empirical Novelty And Significance:** 3
**Recommendation:** 6

**Clarity, Quality, Novelty And Reproducibility:**

Please see weaknesses for clarification questions.

Please fix cite/citep/citet use.

**Strength And Weaknesses:**

This paper presents an interesting, clear, and effectively simple (in a good way) idea for dataset pruning. Here are a summary of strengths and weaknesses

**Strengths**
1. **Effective empirical results** - The authors demonstrate on CIFAR10 & CIFAR100 the effectiveness of their approach. It would appear to have clear improvements over baseline methods, particularly in ~60% pruning.
2. **Simple and well motivated idea** - The proposed idea of using influence functions to select which training points to use is clear and understandable and well suited for the task of reducing training complexity

**Weaknesses and clarifications**

3. **Motivation for Optimization** - I think it would be helpful to justify the need for the constrained optimization problem, compared to say a greedy approximation -- which would select top $m$ examples by $||\mathbb{S}_i||_2$. This is said in "high-gradient-norm samples could be zero if the direction of these two samples’ gradient is opposite", but how does this compare theoretically? empirically?

4. **Computing Influence Functions (efficiency)** - It would be helpful to clarify the computational cost of computing influence functions in the empirical analysis.

5. **Computing Influence Functions (during training)** - Is there no benefit to incrementally pruning while training?

6. **Methodological Depth** - I support the simplicity of the approach, however, it is not entirely clear to me whether it would meet the expectations of an ICLR conference paper in terms of the depth to which the authors explore the problem at hand. It seems there are many open questions about (1) class imbalance pruning (2) which parameters are used for influence function computation (3) approximations / exact / efficiency/ tradeoffs  of solution of constrained optimization problem?

**Summary Of The Paper:**

**Overall Summary** This paper presents an approach aims to selecting a subset of training data that (1) makes training more computationally efficient while (2) incurring little-to-no loss in accuracy. The proposed approach is analyzed in terms of its generalization performance in addition to computational requirements.

**Methodological Summary** The proposed method uses influence functions to select which datapoints to keep and which to prune in the given training dataset.

**Empirical Summary** The authors perform extensive empirical analysis that compares to baseline methods (herding, forgetting, random), across architectures, and across training settings (standard training, architecture search).

**Summary Of The Review:**

This paper presents a simple and effective method for dataset pruning using influence functions. There are effective empirical results, but questions about depth of methodological contribution and clarity around some technical details of the approach.

---

> ### Author Response · Authors · 2022-11-13
> **To Reviewer F3Cs (1/2)**
>
> Response to Reviewer F3Cs:
>
> Thanks for taking the time to review our paper! We are happy to respond to your comments and questions as below:
>
> ### **1.Motivation for Optimization**
>
> Thanks for your great suggestion! We have updated the manuscript to include more comparison results in Fig.1. The non-optimization variant of our method by selecting examples based on their own $\|| \mathcal{I_\mathrm{param}} \||$ is shown in Fig.1 (denoted by influence-score). We can see that the optimization-based selection process is crucial for empirical performance, especially when the pruning ratio is high. From the theoretical perspective, if we individually select examples based on their$\|| \mathcal{I_\mathrm{param}} \||$, the group effect $\|| \sum_{\hat{z_i} \in  \hat{\mathcal{D}}} \mathcal{I}_{\mathrm{param}}(\hat{z_i}) \||_2$ in Equation.12 cannot be constrained, and the generalization ability of the pruned dataset cannot be guaranteed.
>
>
> ### **2.Computing Influence Functions (efficiency)**
> The time consumption of computing the influence function is shown below. Since the influence function relies on the assumption that the loss function is strictly convex, we only compute the influence function in Eq.2 for the last linear layer to guarantee the estimation is accurate.
>
> | |CIFAR10 |CIFAR100| TinyImageNet|
> |  ----  | ----  | ----  | ----  |
> | Influence Function (hours)|  3.3 | 3.5 | 6.7 |
>
>
> ### **3.Computing Influence Functions (during training)**
>
> A major advantage of our proposed optimization-based sample-selection method is that we are the first one that can guarantee the 'group effect' (the effect of removing a group of data points). This advantage also brings a limitation to our method, that is, our method cannot select examples on-the-fly (pruning during training). Our method needs to first go through the entire dataset to compute a vector (estimated parameter change) for each data point, then prune the dataset by a designed discrete optimization process so as to guarantee the 'joint influence' of removing a set of examples.
>
> Considering the group effect of removing the combination of arbitrary examples is rational and effective, but also time-consuming, compared with previous score-based methods. But fortunately, dataset pruning only needs to be performed once, and the pruned dataset can be used for subsequent unlimited rounds of model training, parameter tuning, model architecture search, etc., to reduce computational and storage consumption. More efficient dataset pruning is also a future research direction.
>
>
> ### **4.Methodological Depth**
>
> #### **(1) class imbalance pruning**
>
> Thanks for your great suggestion! Validating the dataset pruning on an imbalanced dataset is an interesting idea. We conduct experiments on long-tailed CIFAR-100 with an imbalanced ratio of 100. We compare our method with random selection and class-wise random selection. The results are reported in the form of (head class accuracy/tail class accuracy) in the following table. As illustrated in the results, head classes have more data than tail classes, therefore random pruning would select more head data, leading to the performance drop in head classes. The class-wise random selection prune data randomly in a class-wise manner, however, this would exacerbate the class imbalance issue, resulting in a sharp drop in tail class performance. Our method maintains the performance of both head and tail classes. Note that the results were obtained by using vanilla CE loss without adapting any long-tail data training tricks. We will include more results and discussions in the revised paper.
>
> | Pruning ratio | 0|10|20|
> |  ----  | ----  | ----  | ----  |
> | random selection|  (65.5/7.4) | (63.6/7.6) | (62.3/7.8) |
> | random selection (class-wise)|  (65.5/7.4) | (63.9/5.3) | (62.5/3.9) |
> | ours |  (65.5/7.4) | (65.3/7.3) |(64.9/7.7) |
>
> #### **(2) which parameters are used for influence function computation**
>
> The influence function [1] relies on the assumption that the loss function is strictly convex. To guarantee an accurate estimation of the parameter influence in Eq.2, we fix the extracted features and only compute the parameter influence for the last linear layer, as did in [1].
>
> [1] Koh, Pang Wei, and Percy Liang. "Understanding black-box predictions via influence functions." International conference on machine learning, 2017.

---

> > ### Author Response · Authors · 2022-11-13
> > **To Reviewer F3Cs (2/2)**
> >
> > #### **(3) approximations / exact / efficiency/ tradeoffs of solution of the constrained optimization problem?**
> >
> > Thanks for your great question! The discrete optimization problem in Eq.4 is an NP-hard problem, and it is very time-consuming to get the 'exact solution'. Considering the computational efficiency, we solve the optimization problem in Eq.4 using the simulated annealing algorithm. Simulated annealing is an efficient algorithm that can find an approximated minima in a given time budget. From the table, we can see that the performance of simulated annealing is a good trade-off between the optimal performance (exact solution) and the time consumption.
> >
> > CIFAR10:
> > | Pruning ratio |  10|20| 30| 40| 50|
> > |  ----  | ----  | ----  | ----  | ----  | ----  |
> > | Exact solution (branch and bound) | 95.31(3mins) | 95.08(62mins) |94.36(193mins)|N/A | N/A |
> > | Simulated annealing  | 95.30(30mins)| 95.10(30mins)| 94.23(30mins)| 92.81(30mins)| 88.20(30mins)|
> > | Simulated annealing  | 95.33(5mins)| 95.03(5mins)| 94.22(5mins)| 92.86(5mins)| 87.50(5mins) |
> > | Simulated annealing  |93.66(1mins)| 93.19(1mins)| 92.81(1mins)| 89.50(1mins)| 84.33(1mins) |
> >
> > TinyImageNet:
> > | Pruning ratio |  10|20| 30| 40| 50|
> > |  ----  | ----  | ----  | ----  | ----  | ----  |
> > | Simulated annealing  | 50.8(60mins)| 49.3(60mins)| 46.7(60mins)| 43.6(60mins)| 40.5(60mins)|
> > | Simulated annealing  | 50.7(30mins)| 49.3(30mins)| 46.2(30mins)| 42.5(30mins)| 39.5(30mins)|
> > | Simulated annealing  | 49.3(5mins)| 47.6(5mins)| 45.4(5mins)| 43.1(5mins)| 39.8(5mins)|
> > | Simulated annealing  | 48.6(1mins)| 46.5(1mins)| 44.3(1mins)| 41.2(1mins)| 38.2(1mins)|
> >
> > We will include more details, results, and discussions about the discrete optimization problem in the next revision.
> >
> > We hope our responses have addressed all of your concerns and questions. If there are any further questions, we are very glad to continue the discussion.

---

> ### Author Response · Authors · 2022-11-22
> **Are there any further questions here?**
>
> Dear Reviewer F3Cs,
>
> Thanks a lot for your efforts in reviewing this paper. We tried our best to address the mentioned concerns. Are there any unclear explanations here? We could further clarify them.
>
> Best,
>
> Authors

---

> > ### Comment · Reviewer_F3Cs · 2022-12-07
> > **Thank you so much for the detailed responses**
> >
> > Thank you so much for taking the time to have such detailed responses to my questions. The additional experiments you have added, I believe greatly strengthen the paper. Especially the comparison in "Motivation for Optimization" and "Approximation Tradeoffs". Hence, I've increased my score to a 6.
> >
> > My sincerest apologies for my delayed response to your comments.

---

### Official Review · Reviewer_gH14 · 2022-11-05

**Confidence:** 4
**Correctness:** 4
**Technical Novelty And Significance:** 3
**Empirical Novelty And Significance:** 3
**Recommendation:** 8

**Clarity, Quality, Novelty And Reproducibility:**

The paper is well written and claims are validated. The papers claims appear to be reproducible.

**Strength And Weaknesses:**

Strengths:

+ The paper addresses an important problem given the large datasets in real world, that are often redundant.
+ The validation of the central claim is exhaustive and the paper also evaluates on different architectures.


Weakness

- What is the overall time required for dataset pruning? Do you need to review each data point more than once?



**Summary Of The Paper:**

The paper proposes dataset pruning, a method to remove/reduce training data that has low overall contribution towards model accuracy. The paper uses the concept of influence functions and uses an optimization function to find subsets of data that do not contribute to changes over model parameters. The paper includes theoretical proofs and experiments to validate the claims.




**Summary Of The Review:**

I recommend accepting this paper because it addresses an important problem, has a clear and practical solution and has been empirically validated.

---

> ### Author Response · Authors · 2022-11-13
> **To Reviewer gH14**
>
> Response to Reviewer gH14:
>
> We appreciate the Reviewer gH14 on acknowledging the significance of our paper. Our response to the weakness is as follows:
>
> ### **Q1: What is the overall time required for dataset pruning? Do you need to review each data point more than once?**
>
> A1: Our proposed optimization-based dataset pruning method consists of two stages: (1) compute a vector (using influence function) for each data point, (2) select a sub-training-set that has an overall minimum generalization impact (by a discrete optimization process). The step 1 needs to review each data point for **only once**. The step 2 was conducted on the low-dimensional vector set and it does not need to review the original data set. The time consumption of our dataset pruning method is shown below:
>
> | |CIFAR10 |CIFAR100| TinyImageNet|
> |  ----  | ----  | ----  | ----  |
> | Influence Function (hours)|  3.3 | 3.5 | 6.7 |
> | Optimization (hours)| 0.5 | 1.0 | 1.0|
> |Overall (hours)| 3.8| 4.5| 7.7|
>
> Although dataset pruning, at the current stage, takes a relatively long time (on cifar10, even longer than training a model on full data). But we want to emphasize that (1) Dataset pruning only needs to be performed once, and the pruned dataset can be used for subsequent unlimited rounds of model training, parameter tuning, model architecture search, etc., to reduce computational and storage consumption. (2) Dataset pruning can help the community better understand the generalization ability of a small portion of data. (3) More efficient dataset pruning is also a future research direction.
>
> We hope our responses could address your concern. If there are any further questions, we are very glad to continue the discussion!

---

### Decision · Program_Chairs · 2023-01-20

**Decision:**

Accept: poster

**Justification For Why Not Higher Score:**

- Scalability is a concern
- Need to train a model on entire dataset before pruning
- Not much methodological development
- Evaluation only carried out on simple datasets like CIFAR10 and CIFAR100

**Justification For Why Not Lower Score:**

- Beauty in simplicity which works empirically on CIFAR10 and CIFAR100
- Generalize from dataset selected on small network to larger network as well architecture

**Metareview: Summary, Strengths And Weaknesses:**

The paper attempts to improve efficiency of training machine learning models. In this regards, the authors adopt the path of selecting a subset of dataset and propose a discrete optimization using influence functions to select which datapoints to keep and which to prune in the given training dataset. The authors perform empirical analysis that compares to baseline methods (herding, forgetting, random) on CIFAR10 and CIFAR100 datasets. The reviewers appreciated the simplicity of the approach and the empirical success. We thank the authors and reviewers for engaging in discussion towards improving the paper and adding more results like performance in presence of class imbalance. Also please add all reviewer feedback like adding motivation for the optimization problem and details on how to solve the optimization in the main paper as provided during the discussion.

**Note From Pc:**

if the above contains the word "oral" or "spotlight" please see: "oral" presentation means -> notable-top-5% and "spotlight" means -> notable-top-25%. As stated in our emails, we are disassociating presentation type from AC recommendations

**Summary Of Ac-Reviewer Meeting:**

N/A